# Wind turbines as new smokestacks: Preserving ruralness and restrictive land-use ordinances across U.S. counties

Inhwan Ko[1], Nives Dolšak[2], Aseem Prakash[3]*

**1** Department of Political Science, University of Nevada, Reno, Nevada, United States of America, **2** School of Marine and Environmental Affairs, University of Washington, Seattle, Washington, United States of America, **3** Department of Political Science, University of Washington, Seattle, Washington, United States of America

* aseem@uw.edu

**Data Availability Statement:** All data has been posted on Harvard dataverse: https://doi.org/10.7910/DVN/2VFK1V.

## Abstract

Renewable energy (RE) facilities provide a global public good of climate mitigation but impose local costs such as landscape disruption and harming the rural character. Because of their land-intensive nature, utility-scale RE facilities tend to be located in rural areas with plentiful and cheap land. In the U.S., about every fourth county (729 of 3,143) has enacted ordinances restricting the siting of RE facilities. Drawing on a novel dataset of county-level restrictions on wind and solar RE facilities for the period 2010-2022, we explore whether, all else equal, levels of ruralness motivate the onset of such restrictions. As the policy literature on problem visibility suggests, we find support for this hypothesis for wind energy facilities only, probably because wind turbines due to their height tend to disrupt the rural landscape and are visible from long distances. We also find that counties are more likely to adopt restrictions for both wind and solar when adjacent counties have enacted them, thereby suggesting a contagion effect in the onset of restrictions. Contrary to the prevalent view on partisanship in climate policy, liberal counties are likely to restrict wind facilities. Our paper points to important sociological and quality-of-life factors that might be impeding the clean energy transition.

## Introduction

In recent years, utility-scale wind and solar energy capacities have surged across the world. The International Energy Agency [1] forecasts that globally for the 2022–2027 period, renewable energy (RE) will account for 90% of the additional generation capacity, and by 2025, it will become the largest source of electricity, surpassing coal.

RE facilities can be constructed in specific locations only. In addition to strong wind flow and plentiful sunshine, they require large land areas [2, 3], although the rise of offshore wind could reduce land demand. The land-intensive nature of utility-scale RE facilities explains its geographical concentration in rural areas where land tends to be plentiful and cheap at least in relation to urban areas [4, 5]. However, this location specificity poses a political problem.

**Funding:** The author(s) received no specific funding for this work.

**Competing interests:** The authors have declared that no competing interests exist.

While RE creates the global public good of climate mitigation [6], RE facilities impose costs on local communities [7, 8]. In particular, RE facilities are viewed as disrupting the rural landscape [9, 10] spoiling rural aesthetics, and impacting property values [11–16]. Consequently, across several countries, rural communities are opposing the siting of such facilities, including in Canada [17, 18], Denmark [19], Germany [20], Greece [21], India [22, 23], South Africa [24], South Korea [25], Spain and Portugal [26], the U.K. [27], and the U.S. [28]. This opposition has important implications for energy policy because it may delay or even stop RE projects [29].

This opposition takes various forms including local protests and land-use ordinances that restrict the siting of RE facilities [25, 30]. In this paper, we focus on land-use restrictions against wind and solar energy facilities that have diffused across about one-quarter of U.S. counties. These restrictions have at least one of the four features: (1) setback requirements from certain man-made or natural objects (e.g., property lines, railroads, highways, lakes, rivers, etc.); (2) noise levels (even for solar facilities due to sound from inverters); (3) size restrictions on generation capacity or land area, and (4) ban or moratorium.

Using an event history analysis for the period 2010–2022, we find the level of ruralness of a county drives the onset of such restrictions but for wind only. This supports previous studies that find wind energy facilities (in relation to solar facilities) have larger visual impacts on surrounding landscapes [21, 31] because they are typically larger in size and stand out due to their height. Our findings hold even when we control for county-level confounding factors including economic performance, partisanship, demography, as well as policy spillovers from neighboring counties.

## Drivers of siting restrictions on wind and solar energy facilities

About 27 countries have at least one provincial or local-level government that restricts wind or solar RE facilities. Multiple factors could drive the onset of such restrictions including local residents' unhappiness with the disruption caused by these facilities to their landscapes. This is where the issue of problem visibility becomes theoretically important.

Problem visibility, or issue visibility, refers to the degree to which a particular problem (or issue) is (physically) visible to the public. Scholars note, for instance, that air pollution has higher problem visibility than water pollution [32, 33]. This is because air pollution is physically visible (think of vehicular pollution or factory smokestacks), while water pollution tends to be partially hidden because citizens typically cannot see pipes discharging wastewater into the oceans, lakes, and rivers. Psychologists note that routine and visual encounters shape public perceptions [34, 35]. Consequently, as people go about their everyday business, they probably encounter smokestacks or vehicles belching smoke more often than wastewater pipes. Thus, citizens are more likely to be concerned about air pollution than water pollution. The implication is that citizen mobilization is higher, and governmental response to address citizen concerns is quicker for visible problems as opposed to less visible problems. We recognize that issues get policy attention when they affect privileged (typically wealthy) communities as opposed to less privileged ones [36]. Indeed, as we discuss below, our model controls for community-level wealth in assessing factors driving the onset of restrictive ordinances.

The above discussion suggests that if a county already has RE facilities, the residents probably see them as they drive around the county every day for work or family chores. This also means that they experience the landscape disruption these facilities cause and might be more motivated to restrict the growth of such facilities in the future. For example, in Germany, critics view wind projects to be causing "Verspargelung der Landschaft" (asparagus-isation of their landscape) [37]. In South Korea, some describe wind turbines as "iron stakes" spiked in

the mountains by Japanese colonialists [38]. A report [39] on public opposition to the 300-megawatt Apex Clean Energy project in Ohio's Crawford County Ohio, quotes Josh Strain, an airline pilot, who opposed wind power because he fears that shadows from turbine blades, could pass over his home and make it feel like an industrial park. It quoted another resident who said, "Part of the reason we enjoy living in a rural community is the outdoors and being able to go out at night and look at the stars."

Based on the above discussion, we hypothesize:

H1: Counties with a higher installed capacity of RE facilities (solar and wind) are more likely to enact restrictive ordinances.

If a problem is visible to citizens and it upsets them, we can expect a higher chance of policy response. Prakash and Potoski [40] provide the example of Delhi, the capital city of India, where the local government is more committed to enforcing air pollution laws as opposed to water pollution laws, although both problems are severe. Broadly, scholars note that governments devote resources to visible policy initiatives as opposed to less visible ones [41]. In the context of climate adaptation, Sowers et al. [42] find that governments place a low priority on investing in low-visibility social engagements to enhance adaptive capacity as opposed to creating hard infrastructure such as water supply projects, desalination projects, canals, and dams. Regarding federal spending on natural disasters, Healy & Malhotra [43] find that voters reward political parties for delivering disaster relief (a visible policy but not for investing in less visible disaster management policies.

In the context of RE facilities, because wind turbines are tall and big in relation to solar, they are more disruptive to the rural landscape [21, 31]. U.S. Department of Energy [44] notes:

"A wind turbine's hub height is the distance from the ground to the middle of the turbine's rotor. The hub height for utility-scale land-based wind turbines has increased 66% since 1998–1999, to about 94 meters (308 feet) in 2021. That's about as tall as the Statue of Liberty! [. . ..] A turbine's rotor diameter, or the width of the circle swept by the rotating blades [. . .] has also grown over the years. [. . .] The average rotor diameter in 2021 was 127.5 meters (418 feet)—longer than a football field."

Moreover, wind facilities tend to be larger than solar in terms of their generating capacity. In the U.S., the average capacity of wind energy facilities is 71.46MW, while that of solar energy facilities is 9.8 MW. Given that the mean capacity of wind turbines is about 2.75 MW, a single facility comprises 25 turbines, on average. It is therefore not surprising that wind facilities can often be visually spotted from a considerable distance, as opposed to solar facilities which are smaller in size and typically ground-mounted. Thus, in counties with more rural landscapes, the perceived landscape disruption of wind energy facilities may be larger than in counties without this characteristic. Therefore, rural counties will be more likely to introduce siting restrictions on wind energy facilities than urban counties. Because solar facilities are less visually disruptive in relation to wind energy facilities, we do not expect the level of ruralness to drive the onset of siting restrictions aimed at solar facilities.

H2: Rural counties are more likely to place siting restrictions on wind energy facilities than urban counties.

H3: Rural counties are no different from urban counties in their likelihood of placing siting restrictions on solar energy facilities.

## Data and variables

We draw on the National Renewable Energy Laboratory [45] database on local ordinances for siting wind and solar projects. While this is the most comprehensive data source on this subject, in some instances it does not report the year in which counties introduced land-use restrictions. In such cases, we retrieved the year of ordinance enactment from the county legal documents. We consider any county with at least one of the four restrictive stipulations as having enacted the ordinance. Among 3,143 counties across 50 U.S. states and the District of Columbia, 560 counties have enacted restrictions on wind facilities, and 315 counties on solar facilities. 146 counties have restrictions on both facilities.

## Independent and control variables

The key independent variables of interest are (1) the existing levels of wind and solar energy capacity and (2) the level of ruralness of the county. Thus, our model includes wind and solar energy capacity (in MW) in the county (logged). The data are from the annual survey conducted by the U.S. Energy Information Administration (EIA), Form EIA-860.

Scholars tend to classify counties as either rural or urban based on the threshold measures of population density or size. Recognizing that ruralness is a continuum, Waldorf and Kim [46] introduced the Index of Relative Rurality (IRR) which ranges from 0 to 1, where counties with a value closer to 1 are considered more rural. The value is an unweighted average of four variables that proxy the ruralness of a county: (1) population size, (2) population density, (3) remoteness (distance from the closest metropolitan area), and (4) build-up area as a percentage of total area. For an easier interpretation of our results, we have rescaled it from 0 to 100 by multiplying the index value by 100. Moreover, we recognize that highly urbanized areas may not be land-available for hosting RE facilities. Thus, their baseline hazard for adopting siting restrictions may be extremely low. Therefore, we exclude the top ten IRR counties in a separate model. The original results hold.

Our model controls for several factors that may independently affect the likelihood of a county enacting the ordinance. First, scholars note that partisanship is closely associated with support for climate policy [47–49] which correlates with support for RE [50]. Hence, we control for Republican vote share in the most recent presidential election. The data is retrieved from the MIT [51].

Second, residents in richer counties have more political power to shape their regulatory environment [52]. Thus, we should expect to find that richer counties are more successful in persuading county policymakers to enact restrictive RE ordinances. On the other hand, rich residents may have pro-climate policy preferences, as per the post-materialism hypothesis [53], and might be more supportive of RE facilities. Therefore, without a theoretical prior about directionality, we control for the county's per capita income (logged) and the unemployment rate as reported by the U.S. Bureau of Economic Analysis (BEA).

Third, domestic migration might also affect local RE politics. Individuals might be relocating to rural areas to enjoy rural landscapes and amenities along with more affordable property values [54]. Because RE facilities might undermine such benefits, migrants could oppose RE facilities [18, 55]. On the other hand, long-term residents might feel more attached to the land and its rural character and therefore have the motivation to oppose new RE facilities. Therefore, without a theoretical prior about directionality, we control for the number of *domestic* migrants (those who lived in a different county in the previous year) per thousand population as reported in the U.S. Census.

Fourth, older populations might be more resistant to landscape disruption and oppose RE facilities. Further, as opinion polls suggest [56], the younger generation is more concerned

about climate issues and thus could be more supportive of RE facilities. Thus, we control for county median age as reported in the U.S. Census.

Fifth, because phasing out fossil fuels is the critical pillar of climate policy, fossil fuel communities are hurt by climate action. The coal industry, in particular, has been in decline because as a fuel for electricity generation, natural gas has substantially replaced coal, and in recent years, the share of wind and solar in electricity generation is rising. This massive change in the fuel used for electricity generation has motivated resistance to climate action in coal communities [57]. Thus, we control whether the county has any working coal mines as reported by the U.S. EIA.

Sixth, state-level factors might play a role at the county level RE politics. County residents might perceive a higher risk that a RE facility might be established if the state has adopted Renewable Energy Portfolio Standards (RPS). Thus, we control whether the state has adopted RPS policies as reported by the National Conferences of State Legislatures [58].

Moreover, opposition to RE might even emanate at the state level. Indeed, as of 2022, 11 states have adopted restrictions on the siting of wind energy facilities (no state has adopted such restrictions for solar). For example, Connecticut has a statewide regulation on setback requirements for facilities beyond a certain threshold. A similar regulation exists in the case of Kentucky. Given that counties may implement restrictions as a response to such state-level restrictions (e.g., translate state-level restrictions into county-level stipulations), we also include them in our analysis. The data is also retrieved from the NREL database.

Finally, counties do not make their policies in isolation. As policy diffusion scholars have pointed out, actors often watch, mimic, and learn from neighboring actors who typically face similar political and ecological challenges [59–61]. Thus, we control for the share of the contiguous counties (within or outside the state) with restrictive ordinances. In addition to the ordinances, the neighborhood effect might work via another mechanism. County residents might have visual contact with RE facilities (and recognize their landscape disruption or the lack thereof) as they cross county lines while doing their everyday business. Thus, we also control for the average wind and solar energy capacity in contiguous counties. Table 1 summarizes the definitions of variables and their sources discussed so far. Data and R code used to create and analyze these variables are available here (https://doi.org/10.7910/DVN/2VFK1V).

## Findings

To explore the onset of county-level restriction aimed at RE facilities, we use an event history analysis with 3,143 U.S. counties and county equivalents for the period 2010–2022. We lag our time-varying covariates by one year to capture local policy-making dynamics. We also include county-level random effects to adjust for unit-fixed unobserved confounders. Table 2 shows the descriptive statistics of all variables used in our analysis.

Table 3 summarizes the results of the model for land-use restrictions on wind ("wind model") and solar RE facilities ("solar model"). We find that the level of wind (solar) installed capacity drives the onset of restrictions on wind (solar) energy facilities (H1 is supported). We also find support for our expectations regarding the relationship between ruralness and the onset of restrictions aimed at wind facilities (H2 is supported). This implies that while residents with first-hand experience of wind and solar facilities will seek to restrict both in the future, the desire to protect ruralness comes into play only for wind facilities which are visually more disruptive than solar facilities (H3 is supported).

How does state-level climate policy influence the onset of county-level restrictions? In particular, might state-level RPS increase the likelihood of RE facilities coming to the county? Indeed, we find that counties in RPS-implementing states are more likely to have restrictions

**Table 1. Variable definitions and data sources.**

| Variable | Definition | Source |
|---|---|---|
| Existing wind capacity | The sum of all wind energy facilities' nameplate capacity | US EIA |
| Existing solar capacity | The sum of all solar energy facilities' nameplate capacity | US EIA |
| Index of relative rurality | The unweighted average of population size, population density, remoteness (distance from the closest metropolitan area), and build-up area as a percentage of total area | Waldorf and Kim (2018) |
| Republican vote share | The share of votes to the Republican candidate in the previous presidential election in total votes | MIT (2023) |
| Per capita income | - | US BEA |
| Unemployment rate | - | US BEA |
| Domestic migrants per thousand population | The number of domestic migrants per thousand population | US Census |
| Median age | - | US Census |
| Coal mines | A value of 1 if a county in a given year has operating coal mines; 0 if otherwise | US EIA |
| State-level RPS | A value of 1 if a county is affiliated with the State with a renewable portfolio standards policy; 0 if otherwise | NCSL |
| State-level renewable restriction | A value of 1 if a county is affiliated with the State with renewable energy siting restriction policies; 0 if otherwise | NREL |
| Contiguous restrictions | The share of contiguous counties with renewable energy siting restrictions in total contiguous counties | NREL |
| Contiguous wind capacity | The average of contiguous counties' existing wind energy capacity | US EIA |
| Contiguous solar capacity | The average of contiguous counties' existing solar energy capacity | US EIA |

All variables are observed in each county year.

on both wind and solar energy facilities. Yet, state-level wind restrictions do not have an impact on county-level wind restrictions.

Consistent with our expectation about spatial diffusion of restrictions, a county is more likely to enact such restrictions on both wind and solar if a higher share of contiguous counties

**Table 2. Descriptive statistics.**

| Variable | Mean (Standard deviation) | Unit |
|---|---|---|
| Existing wind capacity | 23.51 (118.17) | MW |
| Existing solar capacity | 6.48 (68.57) | MW |
| Index of relative rurality | 0.50 (0.10) | Normalized score |
| Republican vote share | 61.52 (15.53) | % |
| Per capita income | 42014.85 (12875.8) | USD |
| Unemployment rate | 6.18 (2.86) | % |
| Domestic migrants per thousand population | -0.56 (11.73) | One migrant per thousand |
| Median age | 41.17 (5.31) | Age |
| Coal mines | 0.04 (0.20) | Binary variable |
| State-level RPS | 0.65 (0.48) | Binary variable |
| State-level renewable restriction | 0.10 (0.30) | Binary variable |
| Contiguous restrictions | 1.55 (7.51) | % |
| Contiguous wind capacity | 26.44 (75.09) | MW |
| Contiguous solar capacity | 7.82 (55.15) | MW |

**Table 3. Results of the model for wind and solar energy restrictions.**

| Outcome | Wind restriction | Solar restriction |
|---|---|---|
| Variables | | |
| Total wind energy capacity, logged | 0.115 (0.021)** | - |
| Total solar energy capacity, logged | - | 0.216 (0.053)** |
| Index of relative rurality (IRR) | 0.014 (0.006)** | -0.001 (0.010) |
| Republican vote share | -0.012 (0.004)** | -0.007 (0.006) |
| Per capita income, logged | 0.750 (0.210)** | 0.153 (0.374) |
| Unemployment rate | -0.176 (0.033)** | -0.009 (0.041) |
| Domestic migrants per thousand population | 0.000 (0.003) | 0.007 (0.007) |
| Median age | -0.012 (0.010) | 0.016 (0.015) |
| Coal mining county | 0.306 (0.245) | 0.306 (0.367) |
| % of contiguous counties with wind restriction | 0.016 (0.003)** | - |
| % of contiguous counties with solar restriction | - | 0.026 (0.006)** |
| Contiguous counties' wind capacity, logged | 0.224 (0.035)** | - |
| Contiguous counties' solar capacity, logged | - | 0.210 (0.053)** |
| State-level RPS | 0.449 (0.131)** | 0.440 (0.170)** |
| State-level wind restriction | 0.072 (0.106) | - |
| # of observations | 36056 | 35885 |
| # of events | 485 | 298 |
| I-likelihood | -3682 | -2305.1 |
| Concordance index | 0.743 (0.011) | 0.985 (0.001) |

Standard errors are reported in parentheses. The number of events is smaller than the actual counts due to missing observations in variables.

\*\*-p<0.01

\*-p<0.05.

has already done so. Also, the likelihood of onset increases as contiguous counties have more wind and solar capacity. Residents often traverse county lines in their daily lives. Thus, if the neighboring county is dotted with wind turbines and solar facilities, there is a higher chance that residents will encounter them and hear about them from their friends, family, and business associates in the neighboring counties.

Partisanship also plays a role in the onset of restrictions but in an unexcepted way. While liberals tend to be pro-climate, we find that liberal counties are more likely to adopt the restriction on wind energy facilities [62] even though we have controlled for per capita income, the presence of coal mines, and several demographic factors. Arguably, conservative voters might view RE facilities in terms of their economic impact. Moreover, liberal opposition to climate action can be found in some other instances. Opposition to many mining projects which are critical to energy transition is often led by liberal groups. For example, Uji et al. [63] report that environmental groups are actively opposing Nevada's Thacker Pass lithium mines because they see this mine to be polluting groundwater and harming flora and fauna. However, whether partisanship may or may not translate into support for local climate projects is an issue of further research.

In an event history model, coefficients themselves are hard to interpret substantively. Therefore, Holtmaat et al. [64] suggest that for their easier interpretation, results can be visualized into a series of counterfactual scenarios to assess how changes in independent variables affect the relative risk of an outcome. For instance, to assess whether wealth levels might affect the "risk" that a county might adopt a wind restriction, we could compare two hypothetical

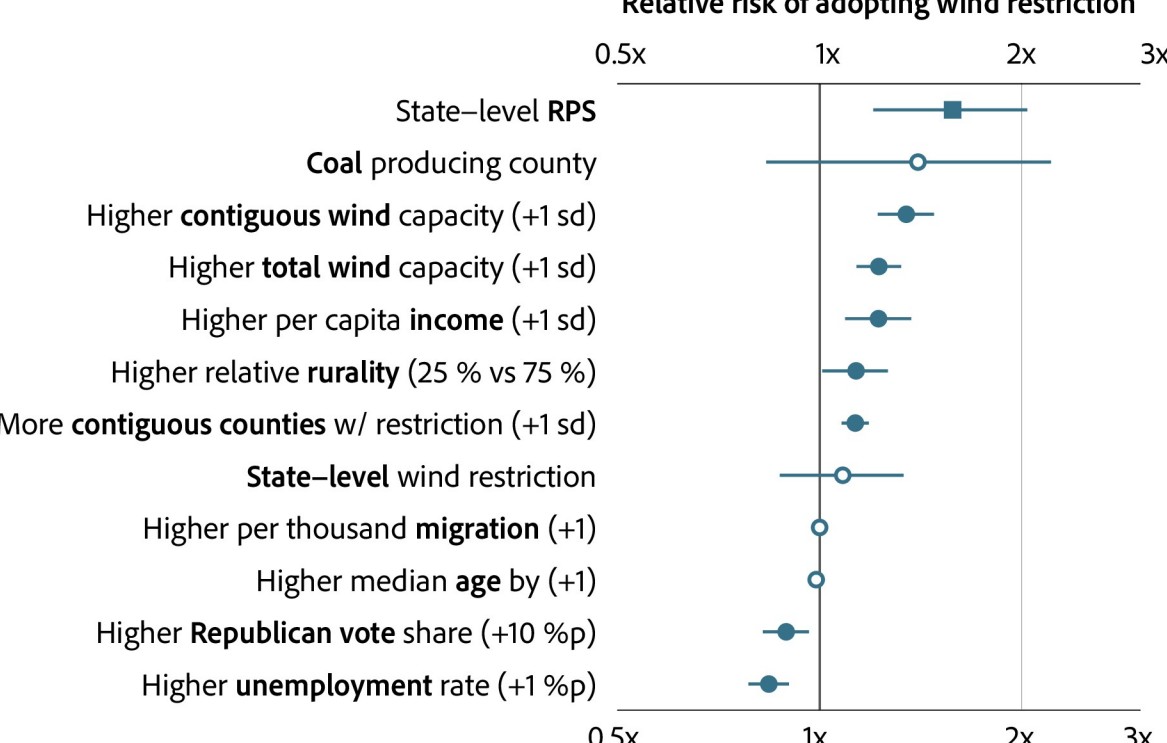

**Fig 1. Drivers of county-level wind restriction.**

counties with all variables held constant, except that one county's per capita income is higher by say some unit.

Fig 1 presents the relative risks of adopting wind restriction in each scenario. A more rural county (which is at the third quartile (75%) on the IRR scale) faces 1.13 times higher risk on average than a more urban county (which is at the first quartile (25%) on the IRR scale) (95% CI: [1.013, 1.26]). A county with an installed wind capacity higher than the average county by 1 standard deviation faces 1.22 times higher risk on average (95% CI: [1.140, 1.315]). Similarly, a county with per capita income higher than the average county by 1 standard deviation faces 1.22 times higher risk of adopting the wind restriction (95% CI: [1.097, 1.361]).

In terms of external drivers, the presence of state-level RPS increases the relative risk by 1.58 times (95% CI: [1.208, 2.028]). Contiguous counties' profile matters: A county faces 1.13 times higher risk of adopting the wind restriction (95% CI: [1.084, 1.177]) when a higher share of contiguous counties by 1 standard deviation adopts restrictions. Also, an increase in the average wind capacity in contiguous counties by 1 standard deviation is associated with a 1.34 times higher risk of a county's adopting the wind restriction (95% CI: [1.227, 1.473]).

Regarding partisanship, a county with a higher Republican vote share by 10% point than the average county faces 0.89 times the risk (that is, lowers the risk levels) of adopting the wind restriction (95% CI: [0.827, 0.958]). Lastly, a county with a higher unemployment rate by 1% point than the average county faces 0.84 times the risk (that is, lowers the risk levels) of adopting the wind restriction (95% CI: [0.787, 0.895]). This might reflect the fact that RE facilities bring new employment and counties having higher unemployment levels are likely to welcome them, as opposed to chasing them away with siting restrictions.

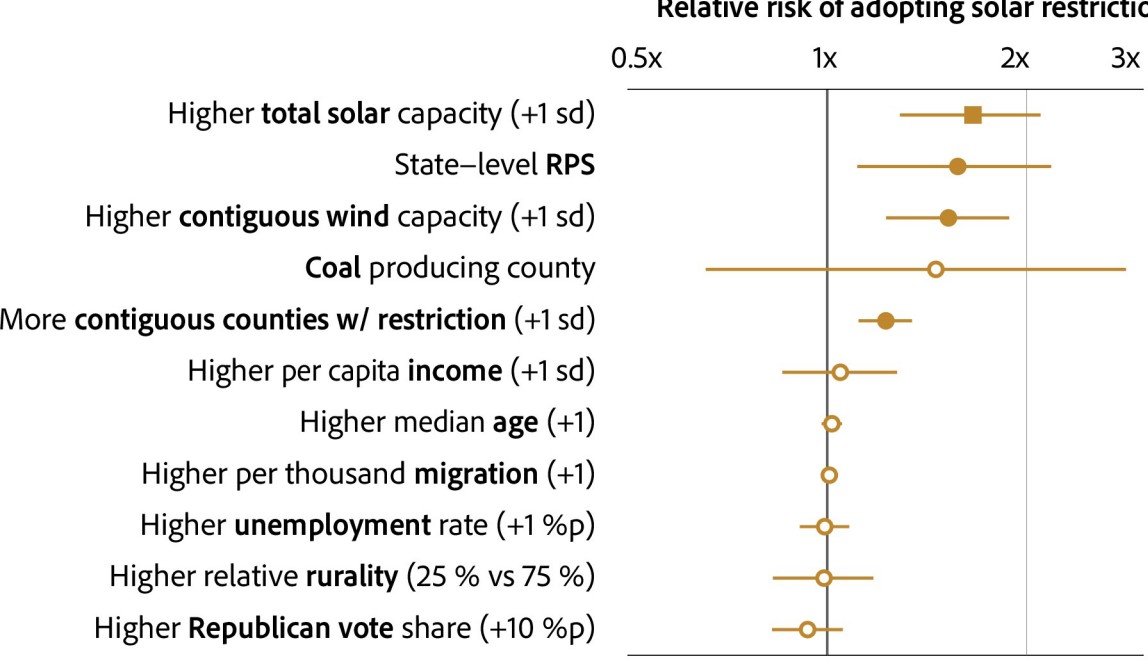

**Fig 2. Drivers of county-level solar restriction.**

Fig 2 presents the relative risks of adopting solar restriction per each scenario. We find that only a handful of factors achieve statistical significance: contiguous counties' restrictions and average solar capacity, state-level RPS, and total solar capacity installed.

## Conclusion

In this paper, we focus on the role of problem visibility in driving the onset of restrictions on RE facilities. We find evidence that while existing levels of wind/solar capacity drive the onset of restrictions for both, the level of ruralness of a county drives the onset of restrictions on wind energy facilities only. Similarly, we find an important role played by neighborhood effects: the number of neighboring counties with restrictions drives restrictions for both and wind. However, the installed capacity of wind in neighboring counties drives the onset of RE restrictions for wind only.

Our findings should also motivate scholars to revisit the relationship between partisanship and support for climate policy. Arguably, our unexpected result about the role of partisanship might reflect the fact that federal climate politics differ from state-level climate politics. After all, Texas leads the nation in RE. In 2022, it generated 136,118 gigawatt-hours from utility-scale wind and solar, well ahead of California with 52,927 [65]. Conservatives might support climate policy as long as it is reframed in non-climate terms. Indeed, Marshall and Burgess [66] report that about one-third of climate policies in U.S. states are passed by Red states, often described in terms of economic development. This may also explain why counties with a higher unemployment rate are less likely to have restrictions on wind energy facilities. Thus, our paper raises important questions about the pros and cons of reframing climate policy in terms of economic development to generate a bipartisan consensus.

Furthermore, we want to emphasize that these county-level restrictions may not necessarily reflect climate denialism. Rather, as the literature suggests, they reflect the desire of rural

residents to protect the integrity and quality of their landscape and rural life. Arguably, there is probably also an element of rural resentment because much of the electricity generated from RE facilities is used by urban residents [67]. In the state of Washington where the local ordinance issue has become salient, Rep. Mark Klicker (R-Walla Walla) who represents a rural area noted that "Eastern Washington communities are burdened with generating RE, while more populated areas west of the Cascades use most of that energy" [68]. Thus, our paper should motivate research into the urban-rural dimension of climate policy instruments.

Our paper raises several questions for future research. Our study is limited to the United States only which has a robust federal structure where local authorities have considerable zooming authority. Local opposition might not translate into restrictions if the zooming authority is vested at different levels of government. Thus, our findings might be idiosyncratic to the U.S. political context.

Second, even in the U.S. context, there is a counter-mobilization at the state level to preempt the authority of county governments to enact such ordinances. New York and Illinois have already passed such laws. And this movement to "ban the bans" is not limited to liberal states. Indiana is a deep Red state and its Republican governor Eric Holcomb is an outspoken supporter of RE which he views in terms of economic development (as opposed to clean energy). For climate supporters, this should be welcome news because it might provide them with some ideas on how to construct bipartisan climate coalitions to speed up the energy transition.

Third the 2022 U.S. Investment Reduction Act (IRA) is pouring vast sums into climate projects. This is shaking up local and state politics and politicians seek to corner funds to create new industries producing car batteries, electric vehicles, solar panels, and wind turbines. Unlike utility-scale wind and solar, there is not much reported opposition to such projects (with the exception of mining). Arguably, as pro-climate projects begin to shape local economies, residents would become more accepting of utility-scale RE facilities and not view them as being imposed by urban environmentalists on rural residents. For example, if a family member works in a factory that produces electric vehicles, arguably this worker's family might become more accepting of RE projects that provide "the fuel" to run electric cars. Thus, future work should examine policy spillover from the IRA to restrictions on utility-scale RE projects, and even new electricity transmission lines which are facing local opposition in many areas.

Finally, we recognize that in the future, there could be new technologies with smaller footprints that might not disrupt the rural landscape. Arguably, offshore wind might fall in this category. But as the controversy in New Jersey offshore wind reveals, local coastal communities, especially the fishing community, believe that offshore wind imposes local costs on them. Yet, we recognize that with radical redesign and technical advances, the issue of landscape disruption might become less relevant.

## Author Contributions

**Conceptualization:** Inhwan Ko, Nives Dolšak, Aseem Prakash.

**Data curation:** Inhwan Ko.

**Formal analysis:** Inhwan Ko.

**Investigation:** Inhwan Ko, Nives Dolšak, Aseem Prakash.

**Methodology:** Inhwan Ko, Nives Dolšak, Aseem Prakash.

**Project administration:** Inhwan Ko, Nives Dolšak, Aseem Prakash.

**Resources:** Inhwan Ko, Nives Dolšak, Aseem Prakash.

**Software:** Inhwan Ko.

**Supervision:** Nives Dolšak, Aseem Prakash.

**Validation:** Inhwan Ko.

**Visualization:** Inhwan Ko.

**Writing – original draft:** Inhwan Ko, Nives Dolšak, Aseem Prakash.

**Writing – review & editing:** Inhwan Ko, Nives Dolšak, Aseem Prakash.

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
