## [Decision Letter · Decision Letter 0]

25 Aug 2023

PONE-D-23-22939Are wind turbines the new smokestacks? Restrictive renewable energy land-use ordinances across U.S. Counties, 2010-2022PLOS ONE

Dear Dr. Prakash,

Thank you for submitting your manuscript to PLOS ONE. After careful consideration, we feel that it has merit but does not fully meet PLOS ONE’s publication criteria as it currently stands. Therefore, we invite you to submit a revised version of the manuscript that addresses the points raised during the review process.

We look forward to receiving your revised manuscript.

Kind regards,

Baogui Xin, Ph.D.

Academic Editor

PLOS ONE

Journal Requirements:

4. We note that Figures 1 and 2 in your submission contain map/satellite images which may be copyrighted. All PLOS content is published under the Creative Commons Attribution License (CC BY 4.0), which means that the manuscript, images, and Supporting Information files will be freely available online, and any third party is permitted to access, download, copy, distribute, and use these materials in any way, even commercially, with proper attribution. For these reasons, we cannot publish previously copyrighted maps or satellite images created using proprietary data, such as Google software (Google Maps, Street View, and Earth). For more information, see our copyright guidelines: http://journals.plos.org/plosone/s/licenses-and-copyright.

a. You may seek permission from the original copyright holder of Figures 1 and 2 to publish the content specifically under the CC BY 4.0 license.  

Additional Editor Comments:

I recommend that it should be revised taking into account the changes requested by the reviewers. Since the requested changes include valuable and constructive reviews, I would like to give you a chance to revise your manuscript. The revised manuscript will undergo the next round of review by same reviewers.

Reviewers' comments:

Reviewer's Responses to Questions

**Comments to the Author**

1. Is the manuscript technically sound, and do the data support the conclusions?

Reviewer #1: Partly

Reviewer #2: Partly

2. Has the statistical analysis been performed appropriately and rigorously? 

Reviewer #1: N/A

Reviewer #2: No

3. Have the authors made all data underlying the findings in their manuscript fully available?

Reviewer #1: Yes

Reviewer #2: Yes

4. Is the manuscript presented in an intelligible fashion and written in standard English?

Reviewer #1: Yes

Reviewer #2: Yes

5. Review Comments to the Author

Reviewer #1: This study is very interesting and provides a new perspective. The language is fluency and easy to read. But still the study has some deficiencies. I hope that my consideration will help to improve the work.

1. It is recommended to revise the abstract and add the innovation and contribution of this study in this section.

2. The format of this manuscripts is not according to PLOS ONE requirement. Please revise it.

3. It is recommended to add a table to show the definitions of all the variables in the research in section 3.

4. Add a table to show the descriptive statistics of all variables in section 4.

5. As for the test of H2 and H3, I think that add interaction term or use subgroup regression is better.

6. This study did not exclude the influence of other factors. For example, the authors said that renewable energy (RE) facilities provide a global public good of climate mitigation but impose local costs such as landscape disruption and harming the rural character, how about if the RE facilities do not impose local costs through technological advances or more perfect design. Please add some robust test.

7. The empirical research surrounding this topic is too incomplete. H1: Counties with a higher installed capacity of RE facilities (solar and wind) are more likely to enact restrictive ordinances.

This can be seen as an inevitable event to some extent. What is truly important in this research process is to study “why”. Please add the mechanism analysis to make the research more meaningful.

Reviewer #2: The paper has an interesting topic, some novel  data and a good structure. Besides the language is fluent and native.  But it has several drawbacks as follows.

First, the variables and the empirical study cannot support the title. Or the author may choose to reconsider the title. As we can see in Section3, the core explainable is the level of ruralness of the country, which cannot clearly proxy any critical factor concerning the title or the hypotheses. According to the hypotheses, there should be variables, which can stand for the restrictive ordinances. But we cannot find any. Also the hypotheses are not closely related to the topic.

Second, the logic of the paper is not clear, especially in Section 2. Hypothesis 2 and Hypothesis 3 cannot necessarily be drawn from their previous argumentation.

Third, the conclusion section is too long as it seems to argue and deduce.

6. PLOS authors have the option to publish the peer review history of their article (what does this mean?). If published, this will include your full peer review and any attached files.

Reviewer #1: No

Reviewer #2: No

---

## [Author Response · Author response to Decision Letter 0]

7 Sep 2023

Response Memo

Wind turbines as new smokestacks: How the desire to preserve ruralness and limit landscape disruption motivates restrictive land-use ordinances across U.S. counties 

(previous title: Are wind turbines the new smokestacks? Restrictive renewable energy land-use ordinances across U.S. counties, 2010-2022)

PONE-D-23-22939

Dear Reviewers: 

Thank you for your excellent and constructive feedback. We are enclosing a memo detailing your suggestions (in italics) and outlining specific ways we address them in the revised manuscript. 

Sincerely,

Authors

Reviewer 1

1. It is recommended to revise the abstract and add the innovation and contribution of this study in this section. 

Response: 

Done.

2. The format of this manuscript is not according to PLOS ONE requirements. Please revise it. 

Response: 

Done.

3. It is recommended to add a table to show the definitions of all the variables in the research in section 3.

Response: 

Done.

4. Add a table to show the descriptive statistics of all variables in section 4.

Response: 

Done.

5. As for the test of H2 and H3, I think that add interaction term or use subgroup regression is better.

Response:

Thank you for this feedback. As we noted in the manuscript, a county’s ruralness is not a binary variable but a continuum. Therefore, we used the index of relative rurality. Creating subgroups would involve drawing an arbitrary distinction between rural and urban counties based on say population size, density, housing stock, or road connectivity. For example, if we take the cutoff as say 5,000 residents (as per the 2020 U.S. census), we will be placing both small urban areas that exist in the midst of rural areas, together with sprawling metropolis such as Los Angeles, whose residents might not view rural landscapes in their everyday lives. Hence, the continuum/index approach to assess ruralness is theoretically more suited to test our hypotheses.

6. This study did not exclude the influence of other factors. For example, the authors said that renewable energy (RE) facilities provide a global public good of climate mitigation but impose local costs such as landscape disruption and harming the rural character, how about if the RE facilities do not impose local costs through technological advances or more perfect design. Please add some robust test.

Response: 

Good point. Arguably, in the future, there could be new technologies with smaller footprints that might not disrupt the rural landscape. Currently, however, we cannot think of any widely adopted RE technology that does not impose local costs. Might offshore wind projects fall in this category? As the controversy in New Jersey offshore wind reveals, local coastal communities, especially the fishing community, believe that offshore wind imposes local costs on them. In the concluding section, we acknowledge that with radical redesign and technical advances, the issue of landscape disruption might become less relevant.

7. The empirical research surrounding this topic is too incomplete. H1: Counties with a higher installed capacity of RE facilities (solar and wind) are more likely to enact restrictive ordinances. This can be seen as an inevitable event to some extent. What is truly important in this research process is to study “why”. Please add the mechanism analysis to make the research more meaningful.

Response: 

Thank you. We have elaborated on the key mechanism, landscape disruption, in the revised paper.

 

Reviewer 2

1. The variables and the empirical study cannot support the title. Or the author may choose to reconsider the title. As we can see in Section 3, the core explainable is the level of ruralness of the country, which cannot clearly proxy any critical factor concerning the title or the hypotheses. According to the hypotheses, there should be variables, which can stand for the restrictive ordinances. But we cannot find any. Also, the hypotheses are not closely related to the topic.

Response: 

Excellent suggestion. The new title is: “Wind turbines as new smokestacks: How the desire to preserve ruralness and limit landscape disruption motivates restrictive land-use ordinances across U.S. counties.”

2. The logic of the paper is not clear, especially in Section 2. Hypothesis 2 and Hypothesis 3 cannot necessarily be drawn from their previous argumentation.

Response:

Fair point, we have streamlined the argument.

3. The conclusion section is too long as it seems to argue and deduce.

Response: 

We have shortened the conclusion.

---

## [Decision Letter · Decision Letter 1]

24 Oct 2023

PONE-D-23-22939R1Wind turbines as new smokestacks: How the desire to preserve ruralness and limit landscape disruption motivates restrictive land-use ordinances across U.S. countiesPLOS ONE

Dear Dr. Prakash,

Thank you for submitting your manuscript to PLOS ONE. After careful consideration, we feel that it has merit but does not fully meet PLOS ONE’s publication criteria as it currently stands. Therefore, we invite you to submit a revised version of the manuscript that addresses the points raised during the review process.

I recommend that it should be revised taking into account the changes requested by the reviewers. Since the requested changes include valuable and constructive reviews, I would like to give you a chance to revise your manuscript. The revised manuscript will undergo the next round of review by same reviewers.

We look forward to receiving your revised manuscript.

Kind regards,

Baogui Xin, Ph.D.

Academic Editor

PLOS ONE

Reviewers' comments:

Reviewer's Responses to Questions

**Comments to the Author**

1. If the authors have adequately addressed your comments raised in a previous round of review and you feel that this manuscript is now acceptable for publication, you may indicate that here to bypass the “Comments to the Author” section, enter your conflict of interest statement in the “Confidential to Editor” section, and submit your "Accept" recommendation.

Reviewer #1: (No Response)

Reviewer #2: (No Response)

2. Is the manuscript technically sound, and do the data support the conclusions?

Reviewer #1: (No Response)

Reviewer #2: Partly

3. Has the statistical analysis been performed appropriately and rigorously? 

Reviewer #1: (No Response)

Reviewer #2: No

4. Have the authors made all data underlying the findings in their manuscript fully available?

Reviewer #1: (No Response)

Reviewer #2: Yes

5. Is the manuscript presented in an intelligible fashion and written in standard English?

Reviewer #1: (No Response)

Reviewer #2: Yes

6. Review Comments to the Author

Reviewer #1: (No Response)

Reviewer #2: The empirical section is sounder, but the logic is still not so clear. For example, how the ruralness is related to the existing level of wind and solar energy capacity.

Still the title is not so proper，not clear and very long.

The authors have not adequately properly addressed the comments raised in a previous round of review.

7. PLOS authors have the option to publish the peer review history of their article (what does this mean?). If published, this will include your full peer review and any attached files.

Reviewer #1: No

Reviewer #2: No

---

## [Author Response · Author response to Decision Letter 1]

26 Oct 2023

Response Memo

Wind turbines as new smokestacks: 

Preserving ruralness and restrictive land-use ordinances across U.S. counties

(Previous title: Wind turbines as new smokestacks: How the desire to preserve ruralness and limit landscape disruption motivates restrictive land-use ordinances across U.S. counties)

PONE-D-23-22939R1

Dear Reviewers: 

Thank you for your comments on how to revise the paper. We enclose a memo detailing your suggestions (in italics) and outline specific ways we address them in the revised manuscript. 

Sincerely,

Authors

Reviewer 2

1. The empirical section is sounder, but the logic is still not so clear. For example, how the ruralness is related to the existing level of wind and solar energy capacity?

Response: 

Thank you for the opportunity to refine our logic. Recent studies provide additional evidence that wind and solar energy facilities are more likely to be located in rural areas (Nilson and Stedman, 2022; O’Shaughnessy et al., 2023). The reason is that these facilities (especially solar) are land-intensive (Denholm et al., 2009; van de Ven et al., 2021), and rural areas have cheaper land. Meanwhile, the existing level of wind and solar energy capacity might increase the likelihood of the onset of land-use restrictions because as their negative impact becomes visible, existing facilities provoke local opposition (Bell et al., 2005; Gross, 2020). Therefore, we control for the existing level of wind and solar energy capacity because they might confound the relationship between the level of ruralness and the likelihood of land-use restrictions, the key variable of interest. 

To refine the logic regarding Hypotheses 2 and 3, we have revised a paragraph on Page 7 as follows:

“Moreover, wind facilities tend to be larger than solar in terms of their generating capacity. In the U.S., the average capacity of wind energy facilities is 71.46MW, while that of solar energy facilities is 9.8 MW. Given that the mean capacity of wind turbines is about 2.75 MW, a single facility comprises 25 turbines, on average. It is therefore not surprising that wind facilities can often be visually spotted from a considerable distance, as opposed to solar facilities which are smaller in size and typically ground-mounted. Thus, in counties with more rural landscapes, the perceived landscape disruption of wind energy facilities may be larger than in counties without this characteristic. Therefore, rural counties will be more likely to introduce siting restrictions on wind energy facilities than urban counties. Because solar facilities are less visually disruptive in relation to wind energy facilities, we do not expect the level of ruralness to drive the onset of siting restrictions aimed at solar facilities.”

H2: Rural counties are more likely to place siting restrictions on wind energy facilities than urban counties. 

H3: Rural counties are no different from urban counties in their likelihood of placing siting restrictions on solar energy facilities. 

Reference:

Bell, D., Gray, T., & Haggett, C. (2005). The ‘social gap’ in wind farm siting decisions: explanations and policy responses. Environmental Politics, 14(4), 460-477.

Denholm P., Hand M., Jackson M, & Ong, S. (2009). Land-use requirements of modern wind power in the United States. National Renewable Energy Laboratory. Available from: https://www.osti.gov/biblio/964608

Gross, S. (2020). Renewables, land use, and local opposition in the United States. Brookings Institution. Available from: 

https://www.brookings.edu/articles/renewables-land-use-and-local-opposition-in-the-united-states/

Nilson, R.S., and Richard C. Stedman (2022). Are big and small solar separate things?: The importance of scale in public support for solar energy development in upstate New York. Energy Research & Social Science 86: 102449. 

O’Shaughnessy E., Wiser R., Hoen B., Rand J. & Elmallah S. (2023). Drivers and energy justice implications of renewable energy project siting in the United States. Journal of Environmental Policy & Planning, 25(3), pp.258-272.

Van de Ven D., Capellan-Peréz I., Arto I., Cazcarro I., de Castro C., Patel P. & Gonzalez-Eguino M. (2021). The potential land requirements and related land use change emissions of solar energy. Scientific Reports, 11(1), p.2907.

2. Still the title is not so proper, not clear and very long.

Response:

The new title is: “Wind turbines as new smokestacks: Preserving ruralness and restrictive land-use ordinances across U.S. counties.”

---

## [Editor Report · Decision Letter 2]

5 Nov 2023

Wind turbines as new smokestacks: Preserving ruralness and restrictive land-use ordinances across U.S. counties

PONE-D-23-22939R2

Dear Dr. Prakash,

We’re pleased to inform you that your manuscript has been judged scientifically suitable for publication and will be formally accepted for publication once it meets all outstanding technical requirements.

Kind regards,

Baogui Xin, Ph.D.

Academic Editor

PLOS ONE
---

## [Editor Report · Acceptance letter]

17 Nov 2023

PONE-D-23-22939R2 

Wind turbines as new smokestacks: Preserving ruralness and restrictive land-use ordinances across U.S. counties 

Dear Dr. Prakash:

I'm pleased to inform you that your manuscript has been deemed suitable for publication in PLOS ONE. Congratulations! Your manuscript is now with our production department. 

Kind regards, 

on behalf of

Professor Baogui Xin 

Academic Editor

PLOS ONE